# N-NOSE Proves Effective for Early Cancer Detection: Real-World Data from Third-Party Medical Institutions

**DOI:** 10.3390/biomedicines12112546

**Published:** 2024-11-07

**Authors:** Nobushige Nakajo, Hideyuki Hatakeyama, Masayo Morishita, Eric di Luccio

**Affiliations:** 1Graduate School of Systems Life Sciences, Kyushu University, 744 motooka, Nishi-ku, Fukuoka 819-0395, Japan; nakajo.nobushige.580@m.kyushu-u.ac.jp; 2Hirotsu Bio Science Inc., 22F The New Otani Garden Court, 4-1 Kioi-cho, Chiyoda-ku, Tokyo 102-0094, Japan; h.hatakeyama@hbio.jp (H.H.); m.morishita@hbio.jp (M.M.)

**Keywords:** N-NOSE, urine-based cancer screening, *Caenorhabditis elegans*, early cancer detection, PET/CT, real-world data

## Abstract

Cancer remains a leading cause of mortality in Japan, yet participation in conventional screening programs is low due to invasiveness, cost, and accessibility. Non-invasive, affordable, and accurate methods for early cancer detection in asymptomatic individuals are urgently needed. This opinion manuscript evaluates nematode cancer testing, a novel urine-based screening test using the nematode *Caenorhabditis elegans*, for its potential to improve early cancer detection rates, based on real-world data published in a Japanese journal. Nematode cancer testing leverages the nematode’s ability to detect cancer-associated volatile compounds in urine, offering a highly sensitive, non-invasive screening approach. We analyzed data from a nationwide survey of PET-equipped medical institutions in Japan, comparing cancer discovery rates between nematode cancer testing-triggered and standard PET screenings. In nematode cancer testing-triggered PET screenings, the cancer discovery rate was 2.96%, significantly higher than the 1.31% observed in standard PET screenings. The apparent positive predictive value (PPV) of nematode cancer testing was calculated at 2.09%; when adjusted for PET/CT sensitivity, the actual PPV increased to approximately 11.7%. This reflects a screening efficiency 14.6 times higher than the general population’s cancer incidence. These findings indicate that nematode cancer testing successfully detects cancer in high-risk individuals and may encourage participation in further diagnostic evaluations. The recently published nationwide survey of PET-equipped medical institutions in Japan highlights the good performance of nematode cancer testing in cancer detection. As an opinion-type manuscript based on real-world data from Japan, this paper shows that nematode cancer testing has substantial potential as a cost-effective, minimally invasive primary cancer screening tool for asymptomatic populations. By enhancing early detection rates and screening efficiency, it addresses the limitations of traditional screening methods. Implementing nematode cancer testing could lead to improved cancer outcomes, particularly in populations with low participation in standard screening programs and resource-limited settings.

## 1. Introduction

Cancer remains a leading cause of mortality in Japan [1], yet participation in conventional screening programs is low due to factors such as invasiveness, cost, and limited accessibility. This underscores the urgent need for non-invasive, affordable, and accurate early detection methods. N-NOSE is a nematode cancer test, a novel urine-based screening test utilizing *Caenorhabditis elegans* (*C. elegans*), that offers a promising alternative. In total, 945,055 new cases were reported in 2020, representing approximately 0.8% of the Japanese population. The most common causes of cancer death in 2022 were lung (76,663 deaths), colon and rectum (53,088 deaths), stomach (40,711 deaths), pancreas (39,468 deaths), and liver (23,620 deaths), which means that half of the Japanese population will experience cancer in their lifetime (62.1% of males and 48.9% of females). Early detection and subsequent intervention can significantly improve healthcare outcomes; however, despite recommendations, half of the individuals eligible for annual screening programs for the five most common cancers (lung, stomach, colon and rectum, breast, and cervix) do not participate [2]. These annual cancer screening programs are conducted by the Ministry of Health, Labour, and Welfare, Japan, as one of the most important national policies to reduce cancer incidence and mortality, and they are categorized as a “population-based screening” for maximizing benefits and minimizing the risk balance of cancer screening in groups. For example, cancer discovery rates based on annual screening programs in the fiscal year of 2015 are as follows: lung (0.05%; 2607 cancers detected in 5,719,736 individuals), stomach (0.12%; 3529 cancers detected in 3,013,168 individuals), colon and rectum (0.22%; 14,968 cancers detected in 6,847,472 individuals), breast (0.33%; 9918 cancers detected in 3,012,808 individuals), and cervix (0.04%; 1628 cancers detected in 4,230,282 individuals) [3] (Table 1). Careful consideration is required because the real-world data of cancer screening tests are sometimes inconsistent with the data from their clinical studies, most probably due to the differences in detailed application procedures between laboratory and social implementation. Moreover, it is noted that the collection of real-world data for newly developed cancer tests is not easy. To address the issue of the low participation rate in annual cancer screening programs, universal, non-invasive, affordable, and accurate methods for early cancer detection, particularly for screening asymptomatic individuals, are urgently needed.

### 1.1. An Overview of Nematode Cancer Testing as an Organism-Based Cancer Screening Method

In recent years, a new type of cancer test called N-NOSE, an organism-based cancer screening method, has garnered attention, with over 700,000 individuals having undergone the cancer test in Japan at the time of this writing. N-NOSE is a novel, non-invasive urine-based screening test that leverages the chemotactic ability of *C. elegans* to detect cancer-associated volatile compounds (VOCs) [6,7,8,9,10,11,12,13,14,15]. Hirotsu et al. first demonstrated that *C. elegans* displays attractive chemotaxis toward secretions from human cancer cells, cancer tissues, and urine from cancer patients while showing an aversive response to urine from healthy subjects [15]. *C. elegans* has an exceptional olfactory system capable of detecting minute concentrations of VOCs associated with cancer cells. This chemotactic response allows the nematode to move toward or away from specific compounds, which can be measured and interpreted as an indicator of cancer presence. By utilizing the nematode’s highly sensitive olfactory system, N-NOSE can detect a broad range of cancers earlier than conventional methods, leading to higher cancer discovery rates in subsequent confirmatory screenings. N-NOSE aims to overcome the barriers of traditional screening methods by providing a simple, non-invasive test that can be administered without the need for hospital visits, potentially increasing participation rates and early detection [6,7,8,9,10,11,12,13,14,15].

Nematode cancer testing is designed to screen asymptomatic individuals to detect cancers early, improving successful treatment and survival chances. Currently available in Japan, N-NOSE implementation relies on “cancer risk” to evaluate each individual’s health and classify it on a scale from A to E, with categories D and E indicating a high risk level. Approximately 5% of N-NOSE examinees fall into the high-risk category [16]. Nematode cancer testing has been evaluated in 30 human clinical studies with various medical institutions domestically in Japan and internationally, all approved by respective ethics committees, resulting in several published research articles [6,10,11,12,13,17,18]. Recently, Hatakeyama et al. reported that *C. elegans* successfully detected over 20 types of cancer with high sensitivities ranging from 60 to 90%, even at earlier pathological stages, in a cohort of over 1600 patients with various cancers [6]. Two independent research groups have also reported the availability and reproducibility of nematode-based cancer detection with a high accuracy for prostate [19] and breast cancer [20]. Galeș et al. mentioned nematode cancer testing and N-NOSE in their review article [21]: “It is important to mention that the N-NOSE behavioral assay has been thoroughly evaluated in clinical studies with large cohorts of patients, showing reliable results. Also, the results have been replicated at other prestigious institutions in the US and Italy”.

The Japanese Society for Biological Diagnosis was founded in 2018 by leading academics and scientists from biotechnology companies with the goal of advancing technology development that leverages living organisms for disease detection. The society focuses particularly on enhancing the accuracy and exploring the future potential of diagnostic tests that utilize living organisms. In addition to the use of trained sniffer dogs for cancer detection [22,23,24,25], Piqueret et al. recently introduced the neglected potential of various invertebrates in detecting disease via olfaction in their review article [26]; four invertebrate species—the nematode *C. elegans*, the fly *Drosophila melanogaster*, the bee *Apis mellifera*, and the ant *Formica fusca*—have already been tested for disease detection. Thus, the potential of non-human animals, including invertebrates, as olfactory biodetectors has been extensively investigated.

### 1.2. The Real-World Data of N-NOSE from Third-Party Medical Institutions

The N-NOSE cancer test has become a milestone in the development of technology that utilizes living organisms for disease detection. In line with this, in September 2024, one Japanese publication provided real-world data on N-NOSE to contextualize its performance nationwide in Japan [5]. The publication is Japanese, limiting its accessibility; nonetheless, it provides valuable insight into the use of living organisms for cancer detection. Nagamachi et al. hypothesized that the PPV of N-NOSE could be estimated by comparing the cancer detection rates in N-NOSE–triggered PET/CT screenings to those in standard PET/CT screenings [5]. Given that the original publication by Nagamachi et al. is not available in English, we have included a detailed summary of their methodology and findings. They calculated the apparent PPV of N-NOSE by assuming PET/CT as the “Standard of Truth (SOT)” for cancer detection. Nagamachi et al. described that for individuals with a high likelihood of cancer, as indicated by N-NOSE, whole-body PET cancer examination would be the next step to identify the types of cancer specifically. They initiated a nationwide retrospective survey from October 2020 to September 2023, on 229 PET-equipped medical institutions, limited to being registered with the PET Nuclear Medicine Subcommittee (Japan). The survey results were returned from 102 institutions out of 229 (45%). In their nationwide survey with 28,175 patients, Nagamachi et al. found that among 778 individuals who underwent N-NOSE–triggered PET screenings, 23 cancers were detected (2.96% detection rate). In contrast, the standard PET screenings detected 359 cancers in 27,397 individuals (1.31% detection rate). These results suggest that N-NOSE effectively identifies individuals at higher risk, leading to a higher yield in subsequent PET screenings. Then, Nagamachi et al. hypothesized the apparent positive predictive value (PPV) of N-NOSE, assuming the PET/CT test as the “Standard of Truth (SOT)”. According to their analysis, the apparent PPV of N-NOSE estimated from examinees with a high likelihood of cancer was 2.09% (22 cancers in 1053 individuals) in all 33 PET institutions where an N-NOSE-triggered PET/CT test was performed.

This survey [5] is informative and suggestive for the following reasons. First, N-NOSE appears to function effectively as a primary cancer screening test for asymptomatic populations without visiting medical institutions, compared to relying solely on whole-body PET cancer screening without prior indication. Second, the apparent PPV of N-NOSE is comparable to those of the five most common cancers in annual screening programs in Japan (stomach: over 1.0%, colon and rectum: over 1.9%, lung: over 1.3%, breast: over 2.5%, and cervix: over 4.0% PPV quality control thresholds) [4] (Table 1). The actual PPV of N-NOSE can also be estimated by considering the sensitivity of PET cancer screening, which has been reported as 17.8% [27], since the PPV of the primary cancer screening test is significantly influenced by the sensitivity of the secondary diagnostic test, and, in general, PET/CT is not a perfect SOT. As clearly described in “The Third edition of The Guideline for FDG-PET Cancer Screening” in 2019 by the Japanese Society of Nuclear Medicine—PET Nuclear Medicine Subcommittee [https://jcpet.jp/cancer-screening/guideline.html (in Japanese), accessed on 20 September 2024], (i) PET exhibits almost no performance to detect specific types of cancer, such as stomach, prostate, etc., and (ii) the evidence for the effectiveness of PET cancer screening is insufficient. Thus, the actual PPV of N-NOSE could be higher than the apparent PPV. Factoring in the sensitivity of PET/CT, the estimated actual PPV of N-NOSE is approximately 11.7% (2.09% × [100/17.8]), which is 5.6 times higher than the apparent PPV. Additionally, the screening efficiency of N-NOSE can be calculated using the following equation:Screening efficiency = Actual N-NOSE PPV (11.7%)/Cancer incidence in the Japanese population (0.8%) = 14.6-fold

The positive predictive value (PPV) of N-NOSE indicates the likelihood that individuals with a positive test result actually have cancer. An adjusted PPV of approximately 11.7% demonstrates a significant improvement over standard screening methods, reflecting higher screening efficiency. Therefore, the implementation of N-NOSE demonstrates excellent performance in effectively identifying individuals suspected of having cancer within the general population based on data from third-party medical institutions [5]. By targeting asymptomatic individuals, N-NOSE has the potential to detect cancers at an earlier stage when there are more options for effective treatments.

### 1.3. The Effectiveness of a Primary Cancer Screening Test

Early detection significantly improves survival rates, as evidenced by the National Cancer Center, Japan’s report on ten-year survival rates improving with earlier-stage diagnoses [28]. We believe that the essential role of a primary cancer screening test is not only to efficiently identify individuals suspected of having cancer within the population but also to promote increased participation in nationwide conventional cancer screening programs. It is widely recognized that Japan has a low participation rate in the annual screening programs for the five most common cancers: lung, stomach, colon and rectum, breast, and cervix, all of which require visits to medical institutions or hospitals. The major reasons why people are reluctant to participate in the programs are as follows: (i) high expenses, (ii) time-consuming and inconvenient (one test for one cancer), (iii) time constraints and limited access (particularly for business professionals and parents with young children), (iv) invasiveness with pain, and additionally (v) self-assessment of their health determines no need to take the programs, which is noteworthy. This fact is truly serious because a large number of asymptomatic people may miss the chance of cancer detection earlier, according to the latest report on “after ten-year survival rates for cancers” from the National Cancer Center, Japan; significantly improved outcomes of “after ten-year survival rates” are observed when patients are diagnosed at the early stages across various types of cancer and treated promptly and appropriately [28]. N-NOSE addresses these barriers by offering a non-invasive, cost-effective, and convenient screening option. Given the challenges with traditional screenings, N-NOSE emerges as a promising alternative. Not only does it simplify the screening process, but it also has the potential to increase participation rates, ultimately leading to earlier detection and better patient outcomes. Low participation in cancer screening is a global issue not limited to Japan. N-NOSE’s non-invasive and cost-effective approach could be adapted worldwide to improve early detection rates in various healthcare settings.

Cancer screening tests at medical institutions generally emphasize specificity higher than sensitivity to ensure that non-cancer cases are correctly identified as non-cancer, minimizing false-positive rates. Indeed, in real-world situations, the sensitivities of the existing cancer screening tests are relatively low. It is important to note that sensitivity and specificity are trade-offs and that PPV—the probability that a positive test truly indicates cancer—is crucial and heavily depends on the disease prevalence of the target group. As a primary cancer screening test, N-NOSE shows a significantly higher sensitivity across multiple cancers, even at early stages.

### 1.4. Methodological Limitations and Biases in the Nationwide PET Survey on N-NOSE

Despite the promising results reported in the nationwide survey of PET-equipped medical institutions in Japan regarding N-NOSE [5], several methodological issues, biases, and scientific flaws limit the validity of the findings [5]. The study relies on a retrospective survey rather than a prospective clinical trial, which undermines its ability to accurately assess N-NOSE’s performance. A low response rate of only 45% and a limited sample size introduce selection bias and reduce the representativeness of the data. The absence of a proper control group and inconsistent risk classification criteria over time further complicate the analysis. Focusing exclusively on PET centers may not reflect the broader population of N-NOSE users, and misinterpretation of the PPV without considering the low prevalence of cancer in the screening population could underestimate the test’s performance. Additionally, the study excludes non-target cancers, fails to acknowledge PET’s limitations, and lacks complete follow-up on cancers that may have been missed by PET but detected by N-NOSE. Scientific and analytical flaws include the absence of statistical analyses to assess significance, failure to consider lead-time bias, incomplete reporting of N-NOSE results (e.g., true positives, false positives), omission of cancer stages at diagnosis, and neglecting verification bias due to the differential follow-up of negative N-NOSE results. These issues collectively raise concerns about the study’s conclusions [5] and highlight the need for a more rigorous method by the authors of this nationwide survey to accurately evaluate N-NOSE’s efficacy.

## 2. Conclusions

Despite the limitation of this nationwide survey, we came to the conclusion that these findings robustly advocate for the potential of the use of living organisms such as nematodes as a revolutionary, cost-effective, and minimally invasive strategy for broad-spectrum early cancer detection in asymptomatic populations. N-NOSE is particularly significant in low- and middle-income countries with limited access to advanced cancer diagnostic methods, potentially contributing to improved outcomes for affected individuals, as discussed by Hatakeyama et al. [6].

## Figures and Tables

**Table 1 biomedicines-12-02546-t001:** Comparison of positive predictive values of N-NOSE and other recommended cancer screening tests for the most common five cancers in Japan.

Cancer Test	Gastric Cancer (Age 40~74)	Colorectal Cancer (Age 40~74)	Lung Cancer (Age 40~74)	Breast Cancer (Age 40~74)	Cervical Cancer (Age 20~74)	N-NOSE (Age 15~)
Methods	Upper GI X-ray	Fecal occult blood	Chest X-ray	Visual palpation and/or mammography	Pap smear	Organism-based urine test
Number of examinees [A]	3,013,168	6,847,472	5,719,736	3,012,808	4,230,282	-
Number of cancers detected [B] ([B]/[A] × 100%)	3529 (0.12)	14,968 (0.22)	2607 (0.05)	9918 (0.33)	1628 (0.04)	22 (-)
Number of persons requiring further medical checkup [C] ([C]/[A] × 100%)	229,421 (7.6)	475,386 (6.9)	94,526 (1.7)	216,541 (7.2)	87,516 (2.1)	-
Number of persons undergoing further medical checkup [D] ([D]/[C] × 100%)	187,498 (81.7)	333,172 (70.1)	78,943 (83.5)	191,112 (88.3)	65,109 (74.4)	1053 (-)
Positive predictive value (PPV) PPV = [B]/[D] × 100%	1.88	4.49	3.30	5.19	2.50	2.09 (apparent) 11.74 (actual)
PPV QC thresholds (%)	>1.0	>1.9	>1.3	>2.5	>4.0	-

The numbers and values are obtained from the report of the National Cancer Center, Japan: Cancer Registry and Statistics, regarding cancer screening [3] and the cancer screening manual [4], and the report Nagamachi et al.: multicenter collaborative study on PET cancer screening and N-NOSE [5]. GI: Gastrointestinal. QC: Quality Control. Statistics details: [A], Number of examinees; [B], Number of cancers detected; [C], Number of persons requiring further medical checkup; [D], Number of persons undergoing further medical checkup.

## Data Availability

No new data were created or analyzed in this study.

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
