# Peer review of "N-NOSE Proves Effective for Early Cancer Detection: Real-World Data from Third-Party Medical Institutions"

_biomedicines, 2024, doi:10.3390/biomedicines12112546_

Round 1
Reviewer 1 Report
Comments and Suggestions for Authors
I have carefully reviewed the manuscript titled "N-NOSE Proves Effective for Early Cancer Detection: Real- 2 World Data from Third-Party Medical Institutions" by Hideyuki Hatakeyama et al. While the topic is of significant interest and the overall approach is sound, I believe that the manuscript requires major revisions before it can be considered for publication in Biomedicines.
The following are the primary areas that need substantial improvement:
Abstract
1. The abstract lacks detailed information about the methodology used in the study. Detailing how N-NOSE works using the nematode C. elegans to detect cancer-related volatile compounds in urine will increase the understanding of the screening method.
2. While the abstract mentions comparing cancer detection rates between standard and N-NOSE PET screening, it does not address why N-NOSE increases cancer detection rates. Including a brief explanation of the mechanism behind the increased efficacy of N-NOSE provides clarity.
3. The abstract mentions the positive predictive value (PPV) of N-NOSE and its comparison with PET/CT sensitivity but lacks an in-depth interpretation of these values. Providing context on how these values ​​affect screening accuracy would strengthen the findings.
Introduction
1. The introduction relies heavily on statistical data, without providing a clear connection to the research topic of early cancer detection using N-NOSE. Consider integrating statistical data more coherently within the study objectives.
2. The introduction lacks a clear transition from general cancer statistics in Japan to a specific focus on N-NOSE for early cancer detection. Introducing N-NOSE and its importance in the introduction can improve the flow and clarity of the narrative.
3. The introduction mentions N-NOSE as a non-invasive, cost-effective and accurate method for early cancer detection, but does not explain how N-NOSE works or the rationale for selecting C. elegans for cancer. Diagnosis provides more detail about N-NOSE and its potential advantages over traditional screening methods, enhancing the introduction.
1.1. An overview of N-NOSE as an organism-based cancer screening test
1. While the text mentions the chemotactic behavior of C. elegans toward cancer-related volatile organic compounds (VOCs), it could benefit from further explanation of how this behavior translates into cancer detection. Providing a clearer explanation of the mechanism by which C. Elegance detect cancer in urine samples, improves understanding.
2. The text refers to Hirotsu et al., Hatakeyama et al., and GaleÈ™ et al. Without providing specific details or context of your studies. The inclusion of brief insights into the key findings of these studies and how they contribute to the validity and reliability of the N-NOSE as a cancer screening tool strengthens this argument.
1.2. The real-world data of N-NOSE from third-party medical institutions
1. The text could benefit from a clearer explanation of the hypothesis development process by Nagamachi et al. A discussion of the rationale behind estimating the apparent positive predictive value (PPV) of N-NOSE based on PET/CT testing as the “standard of truth” enhances understanding.
2. The text presents a considerable amount of data on cancer detection rates, PPV estimates, and screening efficiency without always providing a clear relationship between these measures. Simplifying data presentation and ensuring a coherent narrative flow improves readability.
1.3. The effectiveness of a primary cancer screening test
1. The text contains a significant amount of information and details that may confuse readers. Simplifying language and structuring information in a more concise and organized manner increases readability.
2. Transition text between the reasons for low participation in traditional screening programs, the importance of early detection and the role of N-NOSE without a clear transition. Coherently linking these points to form a simple discussion improves the flow of the text.
1.4. Methodological Limitations and Biases in the Nationwide PET Survey on N-NOSE
1. The text at the beginning of the analysis repeats the reference to the nationwide survey of PET-equipped medical institutions in Japan. To increase readability, consider avoiding redundancy and presenting the review more simply.
Conclusion
1. The conclusion does not adequately address the methodological limitations and scientific flaws highlighted in the study. It presents a confident assertion about the potential of N-NOSE without acknowledging the need for further validation and improvement.
Reviewer 2 Report
Comments and Suggestions for Authors
This paper discusses the efficacy of N-NOSE, a novel urine screening test, in early cancer detection, with a focus on real-world application data from Japan. Cancer is a leading cause of death in Japan, yet many individuals are reluctant to participate in routine screening due to the invasiveness, high cost, and poor accessibility of traditional methods. Consequently, there is an urgent need for a non-invasive, affordable, and accurate method for early cancer detection.N-NOSE utilizes the nematode Caenorhabditis elegans to detect cancer-related volatile compounds in urine, offering a highly sensitive screening approach. The study analyzed data from nationwide PET (positron emission tomography) facilities in Japan, comparing the cancer detection rates between N-NOSE-triggered screenings and standard PET screenings. The results showed that the cancer detection rate for N-NOSE-triggered PET screenings was 2.96%, significantly higher than the 1.31% rate for standard PET screenings. The apparent positive predictive value (PPV) of N-NOSE was 2.09%, which increased to approximately 11.7% after adjustment for PET/CT sensitivity. These findings indicate that N-NOSE successfully detects cancer in high-risk individuals, potentially encouraging more people to undergo subsequent diagnostic evaluations. The paper concludes that N-NOSE, as a cost-effective and minimally invasive primary cancer screening tool, can enhance early detection rates and screening efficiency, particularly in settings with low participation rates and limited resources. By implementing N-NOSE, it is anticipated that cancer outcomes can be improved, especially among those who are unwilling to participate in traditional screening programs. I just have a few questions for this paper.
1, has the N-NOSE method been implicated in the diagnosis for different cancer types and how about the prediction rates between multiple cancer types? The author should have some more information to indicate whether there is bias or not for this method in testing different cancers.
2, The author should describe the difference between the traditional methods and this novel method, for instance, the sampling methods, inspection time, cost, and convenience.
Reviewer 3 Report
Comments and Suggestions for Authors
All authors represent Hirotsu Bio Science, which is the company providing the N-NOSE cancer screening test, which in turn is based on the ability of C. elegans nematodes to detect cancer-associated odorants from urine samples. In the manuscript, there are tens of references to publications of the company, accompanied with only two references to groups outside of Japan, who have been able to confirm that the test works as expected.
The opinion first discusses the unfortunate reluctance of Japanese people to take part in cancer screenings and then focuses on one article (ref. 16) that is not easily available in PubMed or other reliable sources and not even found by Google search. This is very irritating to the reader, who is unable to justify the strengths and weaknesses of that article without being able to read it by oneself. Apparently the original article is available only in Japanese, so it remains questionable why this manuscript for an opinion would be of interest to world-wide readers of Biomedicines. As stated in the conclusion, “Despite the limitation of this nationwide survey, we came to the conclusion that these findings robustly advocate for the potential of N-NOSE as a revolutionary, cost-effective, and minimally invasive strategy for broad-spectrum early cancer detection in asymptomatic populations.” Thus, the whole manuscript sounds more like an advertisement for the company and its product instead of an objective opinion, and also for that reason does not merit to be published in Biomedicines.
Round 2
Reviewer 3 Report
Comments and Suggestions for Authors
The text of the manuscript has now been improved, but I am still not confident that this opinion on a survey published only in Japanese would be of high interest to readers outside Japan. However, as this opinion may encourage similar surveys to be done also elsewhere by researchers not affiliated with the Hirotsu Bio Science company, I leave this matter to the editors to decide. If otherwise acceptable for publication, the authors should carefully check their text after including the tracked changes, in order to mix the old and new text more fluently with each other.